# Directing Min protein patterns with advective bulk flow

**Sabrina Meindlhumer** [1,5], **Fridtjof Brauns** [2,4,5], **Jernej Rudi Finžgar** [2,5], **Jacob Kerssemakers** [1], **Cees Dekker** [1] **& Erwin Frey** [2,3] ✉

The Min proteins constitute the best-studied model system for pattern formation in cell biology. We theoretically predict and experimentally show that the propagation direction of in vitro Min protein patterns can be controlled by a hydrodynamic flow of the bulk solution. We find downstream propagation of Min wave patterns for low MinE:MinD concentration ratios, upstream propagation for large ratios, but multistability of both propagation directions in between. Whereas downstream propagation can be described by a minimal model that disregards MinE conformational switching, upstream propagation can be reproduced by a reduced switch model, where increased MinD bulk concentrations on the upstream side promote protein attachment. Our study demonstrates that a differential flow, where bulk flow advects protein concentrations in the bulk, but not on the surface, can control surface-pattern propagation. This suggests that flow can be used to probe molecular features and to constrain mathematical models for pattern-forming systems.

Pattern formation is a phenomenon observed in widely different contexts from physics to biology. In cell biology and embryology, it has been studied across species in intracellular[1–3] as well as multicellular systems[4–6]. The term broadly refers to the self-organization of molecules based on physicochemical principles, realized by the interplay of complex reaction networks, transport mechanisms, and guiding cues[1,2,4,7–11]. Intracellular pattern formation is known to play important roles in the positioning of protein assemblies, particularly during cell division[1–3,7,8,12–15].

The Min protein system from *E. coli* bacteria is the best-studied model system for intracellular pattern formation. While rich in complexity with all its known and possible interactions within a cell and its biological role prior to Z-ring formation[13,16], it is at the same time intriguingly simple as its core pattern-forming mechanism essentially comes down to the interaction of only two proteins, MinD and MinE. The interaction of these proteins is widely considered the textbook example for a mass-conserving reaction-diffusion system and has become the subject of numerous theoretical and experimental studies[17–26]. In vitro reconstitution is well-established and relies on imaging of fluorescently labeled purified Min protein on supported lipid bilayers[3,13,20].

MinD is an ATPase, which binds ATP in the bulk solution and subsequently, binds the lipid membrane. Once membrane-bound, it recruits more MinD-ATP, leading to a positive feedback loop with an enhanced binding of MinD-ATP in its vicinity. This process is constantly counteracted by MinE, an ATPase-activating protein that also gets recruited by membrane-bound MinD-ATP. Membrane-bound MinE triggers MinD to hydrolyze its ATP and to detach from the membrane. Back in bulk, MinD exchanges ADP for ATP and starts the cycle anew[1,7,13]. This simplified description (Fig. 1C) is complemented and modified by countless details within the process, such as multimerization[24], the local MinE:MinD stoichiometry[25,27], the formation of a depletion zone[27], bulk-surface coupling[18], and (particularly notable for our study) the so-called MinE switch[22]. The latter describes the ability of MinE to temporarily adopt a latent, non-reactive state upon membrane detachment. Non-switching mutants of MinE that cannot access this latent state were found to still be capable of pattern formation, albeit only within an extremely reduced concentration

[1]Department of Bionanoscience, Kavli Institute of Nanoscience Delft, Delft University of Technology, Delft, the Netherlands. [2]Arnold Sommerfeld Center for Theoretical Physics and Center for NanoScience, Department of Physics, Ludwig-Maximilians-Universität München, Munich, Germany. [3]Max Planck School Matter to Life, Hofgartenstraße 8, 80539 Munich, Germany. [4]Present address: Kavli Institute for Theoretical Physics, University of California Santa Barbara, Santa Barbara, CA 93106, USA. [5]These authors contributed equally: Sabrina Meindlhumer, Fridtjof Brauns, Jernej Rudi Finžgar. ✉e-mail: frey@lmu.de

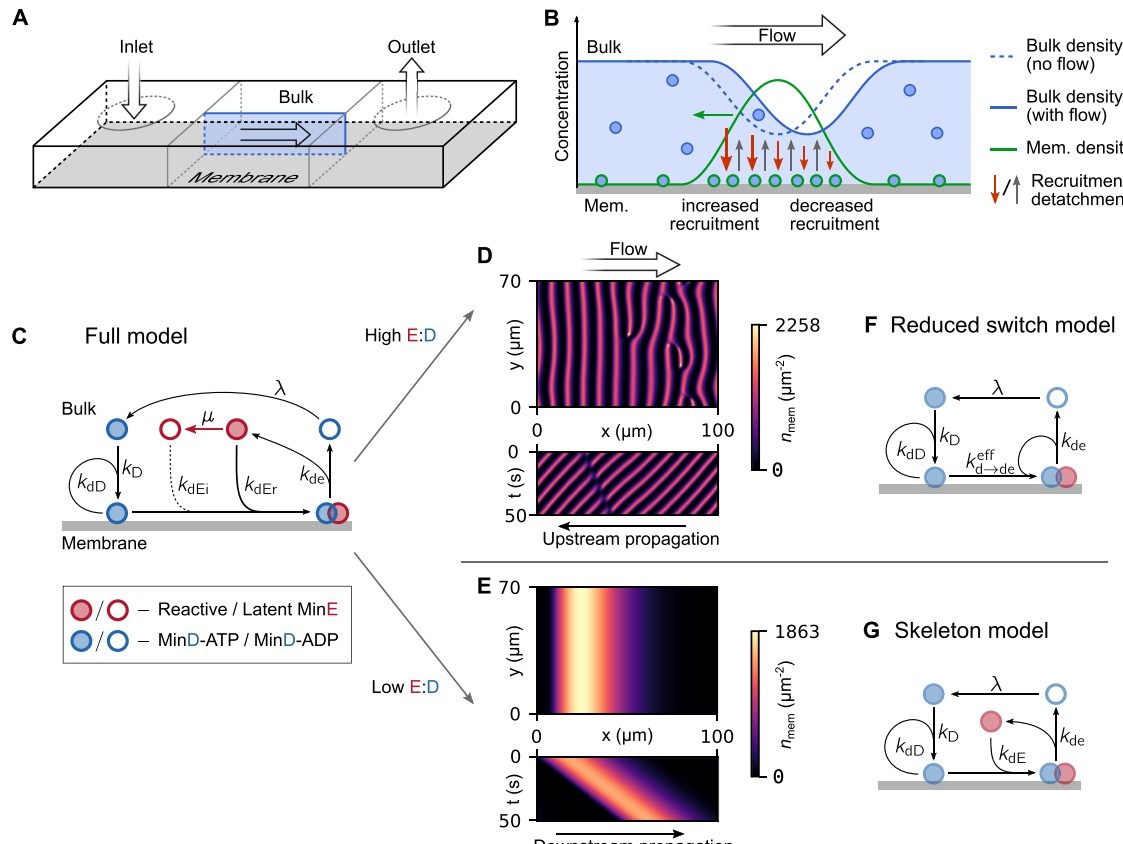

**Fig. 1 | Min models and simulation results. A** Basic illustration of setup used for simulations as well as experiments. **B** Illustration of the effect of bulk flow on pattern formation. MinD advection in the bulk shifts its concentration profile in the bulk relative to the membrane pattern, leading to an increase in the bulk concentration on the upstream side of wave crests relative to the downstream side. This enhances the recruitment rate (green arrows) on the upstream side relative to the downstream side and thus, results in a movement of the membrane pattern (but not the individual proteins) in the upstream direction. **C** Diagram depicting the interactions in the full switch model. This model includes the MinE switch. MinD-ATP binds the membrane, recruiting more MinD-ATP as well as reactive MinE. After MinE stimulates ATP-hydrolysis, MinD-ADP and MinE detach from the membrane. MinD needs to ADP for ATP in the bulk. MinE temporarily assumes a latent state before rebinding to the membrane. **D, E** Typical spatial pattern (snapshots) and kymograph of the membrane protein density found in the full switch model at high E:D ratios (**D**) and at low E:D ratio (**E**). **F** For high E:D ratios the full switch model can be simplified into the reduced switch model. MinE bulk gradients become negligible. **G** At low E:D ratios, the behavior of the Min system is captured well by the skeleton model. This model does not include the MinE switch.

range. The presence of the MinE switch thus increases the robustness of the Min system to concentration fluctuations[22]. Numerous strategies have been employed to study, manipulate, and take advantage of the properties of Min patterns[28]. Examples include changing the membrane or buffer composition[23], crafting surface topology[29], microfabrication of sample chambers[30], variation of sample chamber geometry[18,26], exploration of cargo molecule transport[31], integration with photoswitchable compounds[32], liposome encapsulation[33], and de novo synthesis within liposomes[34].

In an in vitro study by Vecchiarelli et al[23], external hydrodynamic flow was shown to influence Min protein patterns. More specifically, Min protein patterns were observed to propagate *upstream* under fast bulk flow, meaning that they formed waves that were traveling against the direction of the hydrodynamic flow. The authors hypothesized that the cause of this upstream propagation was advective transport of reactive MinE in the vicinity of the membrane—a hypothesis which, to the best of our knowledge, has not been tested to date. More generally, an important implication of this finding is that bulk flow, a macroscopic perturbation, can be used to indirectly probe molecular mechanisms. This motivated us to systematically study the effect of bulk flow on pattern formation in the Min protein system. While it does not appear to fulfill a known biological function in the Min pattern-forming process of *E. coli*, fluid flow was indeed found to be

essential for asymmetry and pattern formation in several other organisms[35]. A prominent example is the establishment of the posterior-anterior axis within the monocellular *C. elegans* zygote, in which actomyosin cortical flows were found to transport regulatory PAR proteins[36–38] and to be associated with cytosolic streaming[39]. A generic consequence of advective flows is the directed transport of proteins. To investigate the effect of such advective transport on intracellular pattern formation in a controlled setting, we here focus on a system that allows for in vitro reconstitution and controlled manipulation of system parameters.

In the investigations presented here, we combine numerical simulations of theoretical models and in vitro experimental analysis to study the influence of bulk fluid flow on Min protein patterns. We find that in response to applied flow, Min protein surface patterns tend to align in mostly planar wave fronts which propagate in the direction of the bulk flow. Furthermore, we noticed that observables such as their preferred direction of propagation can be linked to underlying molecular mechanisms. In this paper, we aim to demonstrate that advective bulk flow can serve as a tool which helps to indirectly probe and reveal these mechanisms. Mathematical models play a key role in this approach as they help to link the molecular mechanisms to the macroscopic observations (e.g. response of protein pattern to bulk flow).

## Results

### MinE-to-MinD concentration ratio determines the propagation direction

Our primary goal is to study the qualitative response of Min patterns to external fluid flow. For simplicity, we consider uniform laminar flow and use a previously established, parsimonious model[8,17,18,22,40] for the Min reaction kinetics as depicted in Fig. 1C. This model describes the basic interactions between MinD and MinE using mass-action law kinetics. MinD binds to the membrane, with a rate that is enhanced by MinD already on the membrane (self-recruitment). Membrane-bound MinD then recruits MinE to the membrane, forming a MinDE complex. In this complex, MinE catalyzes MinD hydrolysis, leading to the dissociation of the complex from the membrane, releasing both constituents into the bulk. In the bulk, MinD undergoes nucleotide exchange before it can bind to the membrane again. The interplay between these basic reactions and diffusive transport gives rise to a rich variety of concentration patterns that form on the membrane, where diffusion is much slower than in the bulk[8,17,18]. Here, we extend this model by accounting for advective transport in the bulk. Membrane-bound proteins are not affected by the flow (see SI). Moreover, since a previous experimental study hypothesized that switching of MinE between reactive and latent states in the bulk is responsible for the upstream propagation of Min patterns[23], we explicitly include this conformational switching of MinE in our model, which we accordingly refer to as the *full (switch) model*[22]. The model equations are provided in the Materials and Methods section with further details in the SI.

We performed finite element simulations in a rectangular area representing the lipid bilayer membrane and the bulk solution above it, choosing periodic boundary conditions in the lateral directions to reduce finite-size effects. The dimension orthogonal to the membrane was integrated out (and explicitly accounting for this dimension does not change the qualitative findings, see SI). We performed simulations for different E:D ratios because previous studies did show that the total concentrations of MinD and MinE, and in particular their E:D ratio, are essential control parameters for Min protein pattern formation[27].

As illustrated in Fig. 1D and Fig. 1E, our simulations show that uniform flow has two main effects: (i) Wave fronts align perpendicular to flow direction and (ii) the wave propagation direction aligns upstream (against the flow) or downstream (with the flow), depending on the E:D ratio. While upstream propagation occurs for high E:D ratios (Supplementary Movie 1), downstream propagation is found for low E:D ratios (Supplementary Movie 2). Notably, we predict downstream propagation in regimes that would not allow for pattern formation in the absence of flow. We refer to this phenomenon as a *flow-driven instability*[41,42]. (Please see SI Sec. 1.7.) Moreover, we observed that downstream propagating patterns slowly increase in wavelength, in a process reminiscent of coarsening dynamics in phase-separating systems. Eventually, only a single propagating soliton-like pulse remained in the simulation domain, as shown in Fig. 1E.

To test our theoretical predictions, we performed experiments with purified Min proteins in flow channels that were coated with a lipid bilayer. To reliably determine the patterns' response to flow, we developed automated tools that allowed us to quantify the propagation speed and direction of wave crests[43]. Figure 2 shows exemplary images along with the results from this wave crest velocity analysis visualized as 2D histograms. Data were collected from multiple comparably sized imaging regions within one flow cell. We experimentally found that an applied advective flow had multiple effects on the Min patterns. We observed a clear decrease in the occurrence of spiral patterns, as patterns tended to transition into traveling waves with wave fronts aligned orthogonal to the flow direction. Importantly, the traveling waves that formed during an applied flow, exhibited *upstream* propagation for high E:D ratio (E:D = 10, Fig. 2A and Supplementary Movie 3), but *downstream* propagation for low E:D ratio

(E:D = 2, Fig. 2B and Supplementary Movie 4). Notably, in a control experiment we reversed the flow rate and found that the pattern's propagation direction also reversed after several minutes (see Fig. S17 and Supplementary Movie 3).

All these experimental observations are in good qualitative agreement with the simulation results. Most importantly, we predicted from our simulations that the relative concentration of MinE with respect to MinD would lead to different outcomes with respect to the patterns' directionality relative to the external flow. There are, however, also notable differences, both qualitative and quantitative. An example of the former is that downstream waves are experimentally observed for E:D ratios that also exhibit pattern formation without any applied flow. This finding stands in contrast to our simulations, where downstream propagation only appears in regimes that would not allow for pattern formation in the absence of flow. Further, we do not observe the predicted coarsening (i.e., a strong increase of wavelength) for downstream propagating waves in our experiments (Fig. S19).

A quantitative difference is the value for the E:D ratio above which upstream propagation can be observed. In experiments, we had to go to much higher E:D ratios (>2) than in the simulations to get upstream propagation, where we observed it starting from E:D ratios of 0.1 (Fig. 3).

### Model reductions and mechanistic explanation of upstream propagation

Reducing a model to the key features dominating within a given parameter regime is a strategy that can provide important insights into the mechanisms underlying the entire process described by the model. To gain intuition about the origin of the E:D dependence of a pattern's response to bulk flow, we studied two reduced models that reproduce the results of the full model in the limits of high and low E:D ratio, respectively.

In the limit of large MinE concentration and fast switching of MinE between the reactive and latent conformations, the MinE switching dynamics can be eliminated using a quasi-steady state approximation (see SI). The resulting *reduced switch model*, visualized by the network cartoon in Fig. 1F, exclusively exhibits upstream propagation in response to flow (Fig. S3). This is consistent with the numerical simulations of the full model in the regime of large E:D ratio (Fig. 3). In the limit where the reduced switch model is valid, bulk concentration gradients of MinE are negligible such that MinE bulk concentrations no longer appear explicitly as dynamic variables. This in turn implies that advective transport of MinE has no effect on the dynamics. As an additional test, we performed simulations of the full model where MinE is not advected by the flow. Consistent with our expectation from the theoretical analysis, we found upstream propagation. Taking the above results together, we conclude that—contrary to an earlier hypothesis[23]—upstream propagation of Min-protein patterns is not due to downstream transport of (reactive) MinE.

This naturally raises the question of what the actual cause of upstream propagation is. To understand this, we consider an incipient accumulation zone of MinD on the membrane. Recruitment of MinD to the membrane acts as a sink, such that the neighboring bulk region becomes depleted of MinD (see Fig. 1B and Fig. S4). This depletion zone is replenished by diffusion, and at the same time transported downstream by the bulk flow. This downstream transport accelerates the replenishment on the upstream side of the accumulation zone and thus, allows faster recruitment of MinD there. Vice versa, recruitment on the downstream side is reduced. As a net effect, one obtains an upstream movement of the accumulation zone (while the individual proteins do not move laterally). This differential flow-induced propagation has been theoretically studied for two-component mass-conserving reaction diffusion models[44]. There, it was shown that flow

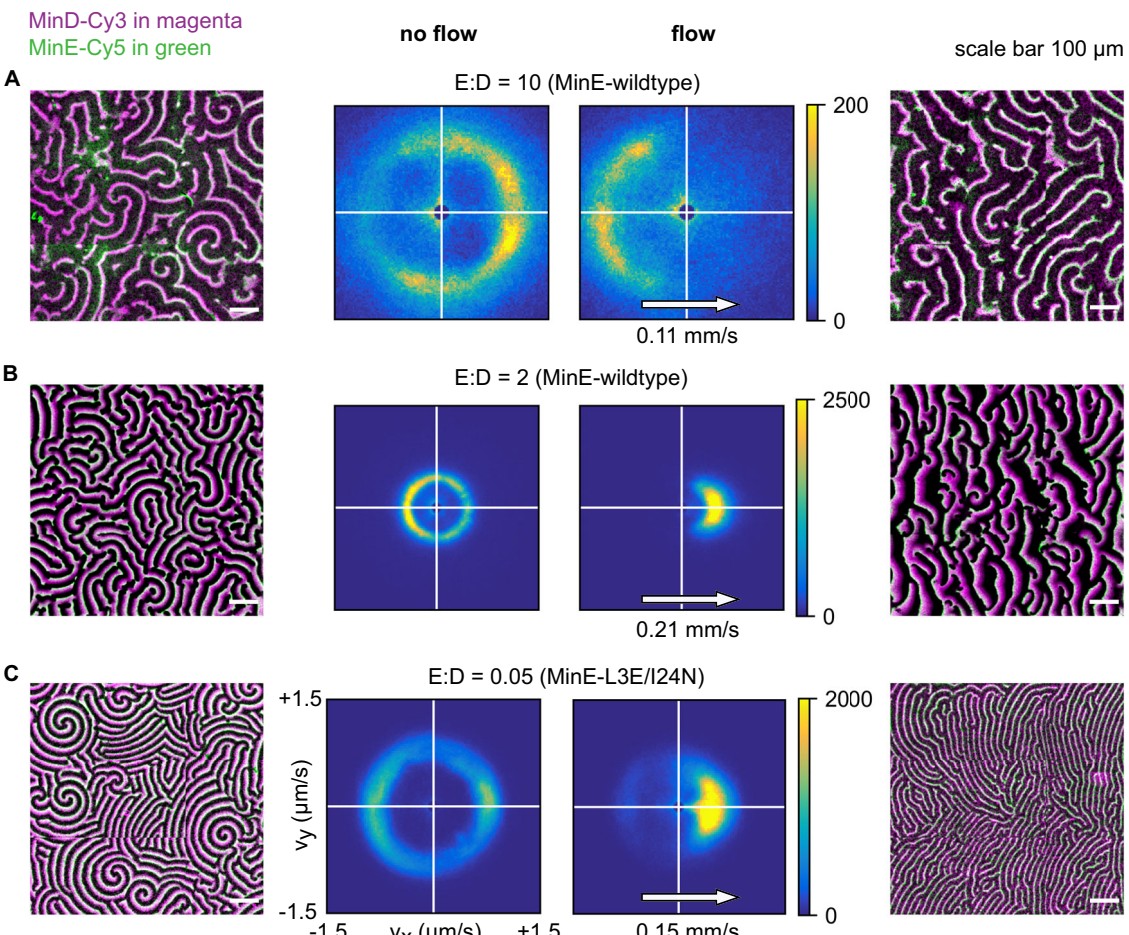

**Fig. 2 | Experimental data showing how patterns respond to flow at different E:D ratios.** Data are for MinE-wildtype (**A**, **B**) and MinE-L3E/I24N (**C**), with bulk flow directed left-to-right. Min patterns (outer left and outer right columns) show MinD-Cy3 in magenta and MinE-Cy5 in green. All scale bars are 100 μm. The results of wave-propagation analysis calculated for MinD-Cy3 data are represented as 2D histograms (center columns) with binning size (25 nm/s) × (25 nm/s), showing counts for directionality ($v_x$, $v_y$). Left half of the figures displays an exemplary image as well as wave-propagation analysis for the no-flow case. Right half of the figures displays an exemplary image as well as wave-propagation analysis with flow. Images were stitched from 3 × 3 fields of view. **A** Upstream propagation was observed in experiments for high E:D ratio (initial 10). **B** Downstream propagation observed for low E:D ratio (initial 2, corrected 1.3). **C** Downstream propagation observed for the MinE-L3E/I24N mutant at E:D = 0.05.

drives the upstream propagation of patterns that are stationary in the absence of flow.

Let us now turn to the regime of low E:D concentration ratio in which we observe *downstream* propagation of the membrane-bound protein waves. Previous studies showed that conformational switching of MinE can be neglected in this regime[18]. This is because the majority of MinE (within the penetration depth of bulk gradients orthogonal to the membrane) is in the reactive form and rapidly cycles between membrane and bulk. Indeed, in simulations of a reduced *skeleton model* that does not include MinE conformational switching, we exclusively found downstream propagating waves, consistent with the notion that the model captures the relevant dynamics in the low E:D regime (Fig. S5). To experimentally test this rationale, we replaced MinE with a non-switching mutant MinE-L3E/I24N[22] and indeed found downstream propagating waves only, as shown in Fig. 2C and Supplementary Movie 5. Notably, we find that pattern formation with this MinE mutant requires sufficiently low E:D ratios in agreement with previous experiments and theory[22].

Next, we tried to decipher the mechanism of downstream propagation. Here, the situation is much more convoluted than in the regime of high E:D, because bulk gradients of both MinD and MinE are significant. While MinD recruits itself to the membrane, MinE recruited by MinD drives MinD detachment by catalyzing MinD hydrolysis.

Intuitively, one might think that the above reasoning for flow-induced upstream propagation might be applied to explain downstream propagation based on MinE advection and the MinE-driven MinD detachment. The reasoning would be that MinE is replenished faster and therefore, recruited faster on the upstream side of the MinD-accumulation zone. This would result in faster MinD detachment there, compared to the downstream side, resulting in a downstream propagation of the MinD-accumulation zone. To test this intuition, we performed simulations in which MinE was not advected by the flow. Strikingly, we *still* observe downstream propagating waves (Fig. S8). This indicates that an intricate interplay of advective MinD transport and diffusive MinE transport is responsible for downstream propagation, whereas MinE advection is not crucial. Disentangling this interplay remains an open challenge for future research.

## Hysteresis and transition to upstream propagation by increasing flow rate

Next, we turned to two closely connected questions: First, how does the propagation direction transition from upstream to downstream (Supplementary Movie 6) at intermediate E:D ratios? Second, how does the flow speed impact the dynamics?

To address these questions, we mapped out a two-dimensional phase diagram employing finite-element simulations using the flow

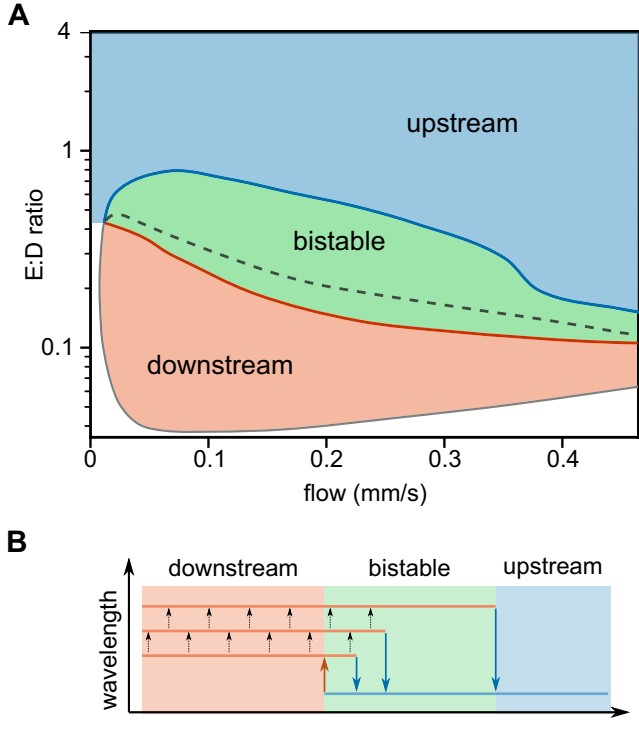

**Fig. 3 | Phase diagram displaying the predicted direction of pattern propagation. A** Phase diagram displaying the predicted direction of pattern propagation. Red and blue regions indicate the parts of the parameter space where exclusively downstream or upstream patterns are observed, respectively. Green region indicates the multistability regime, where the propagation direction depends on the initial conditions. If simulations are initiated from the homogeneous steady state, the observed propagation direction is downstream below the black dashed line, and upstream above it. Details on the adiabatic parameter sweeps and the data points underlying the phase diagram are provided in SI Sec. S1.5 and Fig. S6. **B** Schematic visualizing that the transition flow velocity depends on the wavelength of the pattern. Upon increasing the flow velocity, larger wavelength patterns reverse the propagation direction at a higher flow velocity.

speed and the E:D ratio as control parameters. The resulting phase diagram is shown in Fig. 3A. A striking feature of this phase diagram is that increasing the flow rate can drive a *reversal* from downstream propagation to upstream propagation for intermediate E:D ratios. Importantly, we find that this transition shows hysteresis, meaning that the point on which the transition occurs depends on whether one increases or decreases the flow rate or the E:D ratio. Consequently, there is a regime in which propagation in either direction is possible as the propagation direction sensitively depends on the initial conditions and history. For the downstream to upstream transition, we also observed a correlation with the pattern wavelength (Fig. S6). Downstream propagating patterns were found to slowly coarsen (increase their wavelength) in our simulations. We observed that the downstream-to-upstream transition occurred at higher flow rates for longer wavelength patterns (Fig. 3B).

In a linear stability analysis of the homogeneous steady state, we find two distinct instabilities in the multistable region: One at short wavelengths, corresponding to upstream propagating waves (as indicated by the imaginary part of the growth rate, see SI), and another one at long wavelengths, corresponding to downstream propagating waves. We find that the onset of the first instability (at short wavelengths) precisely coincides with the transition from upstream to downstream propagating waves in simulations with adiabatically decreasing flow velocities. This suggests that upstream propagating waves emerge from this instability. In contrast, we did not find a

characteristic feature in the linear stability properties (encoded in the dispersion relation) that corresponds to the transition from downstream to upstream propagation upon increasing flow rate.

We next tested the predicted hysteresis and multistability of the Min patterns experimentally. Taking advantage of our closed-circle experimental setup, we could incrementally increase the flow rate and acquire protein patterns at distinct points (flow rate, E:D ratio) in the parameter space. Note however, that for practical reasons (such as long incubation/equilibration times), the experimental approach is not identical to the one followed in the simulations depicted in Fig. 3A. In the simulation, we started from a homogeneous steady state at a certain E:D ratio and directly applied a given flow rate. In the experiment, we first established patterns in the absence of flow, and then incrementally went through a sequence of flow rates, with an associated waiting time (15–30 min) at each point.

An example image series of Min patterns at the same location within the flow channel, yet at different bulk flow rates is displayed in Fig. 4A. Analysis of the crest propagation directions as dependent on the bulk flow rate is provided in Fig. 4B–E for different E:D ratios in the intermediate regime. Both upstream and downstream propagating patterns were observed. Figure 4F shows an overview on the peak velocities obtained for different E:D ratios and flow rates. The full crest velocity analysis of all experiments can be found in the SI.

To obtain a quick overview on a pattern's response to flow, we calculated angles from the vectoral components obtained from our wave propagation analysis, binned them in segments of 15° and plotted their normalized occurrence for different flow rates, as shown Fig. 4B–E. This can be understood as a summation over counts found in a certain angular segment from a 2D histogram plot of $(v_x, v_y)$ such as those shown in Fig. 2. In Fig. 4G, we show overviews on results obtained from all experiments done with wildtype MinE. Defining the −30° to +30° segment as "downstream" (shaded red in Fig. 4B–E) and 150° to 210° as "upstream" (shaded blue in Fig. 4B–E), we show the downstream and upstream fractions as red and blue segments respectively, with the symbol radius proportional to the occurrence. Figure S13 provides an overview on the visual representations. At the lowest E:D ratio, downstream propagation was clearly favored upon exposing a pattern to flow, while at the highest E:D ratio, upstream propagation was dominant. For intermediate E:D ratios, the outcome was less clear. Here, we found that the propagation direction sensitively depended on the initial condition, i.e., on the initial propagation direction in the absence of flow.

For most E:D ratios, we observed that waves tended to slow down upon increasing the flow rate (see Fig. 4F and Fig. S18). We were able to confirm that the observed slowing down is indeed induced by the flow and not merely a consequence of the experiment's duration. In a control experiment, the Min pattern in a sample channel was not exposed to flow yet observed over the same time period as a flow-experiment that was run in parallel in a separate flow channel (at an initial E:D ratio of 3). The results of both the regular and control experiment are shown in Figs. S15 and S16. Analysis showed that while the control's pattern did change over time, it did not show the distinctive directional features of the pattern exposed to bulk flow. Slowing down of the wave crest was observed, yet much less pronounced than for the flow experiment. In simulations, we find that waves generally speed up under faster flow, while a slowdown is only observed in a small regime of slow flow speeds (Fig. S9).

## Discussion

In our study, we performed simulations as well as in vitro experiments designed to investigate the influence of advective bulk flow on membrane-bound protein patterns. We theoretically predicted and experimentally showed that Min protein patterns respond differently to hydrodynamic flow depending on the E:D ratio and flow rate. The transition from upstream propagation (high E:D) to downstream

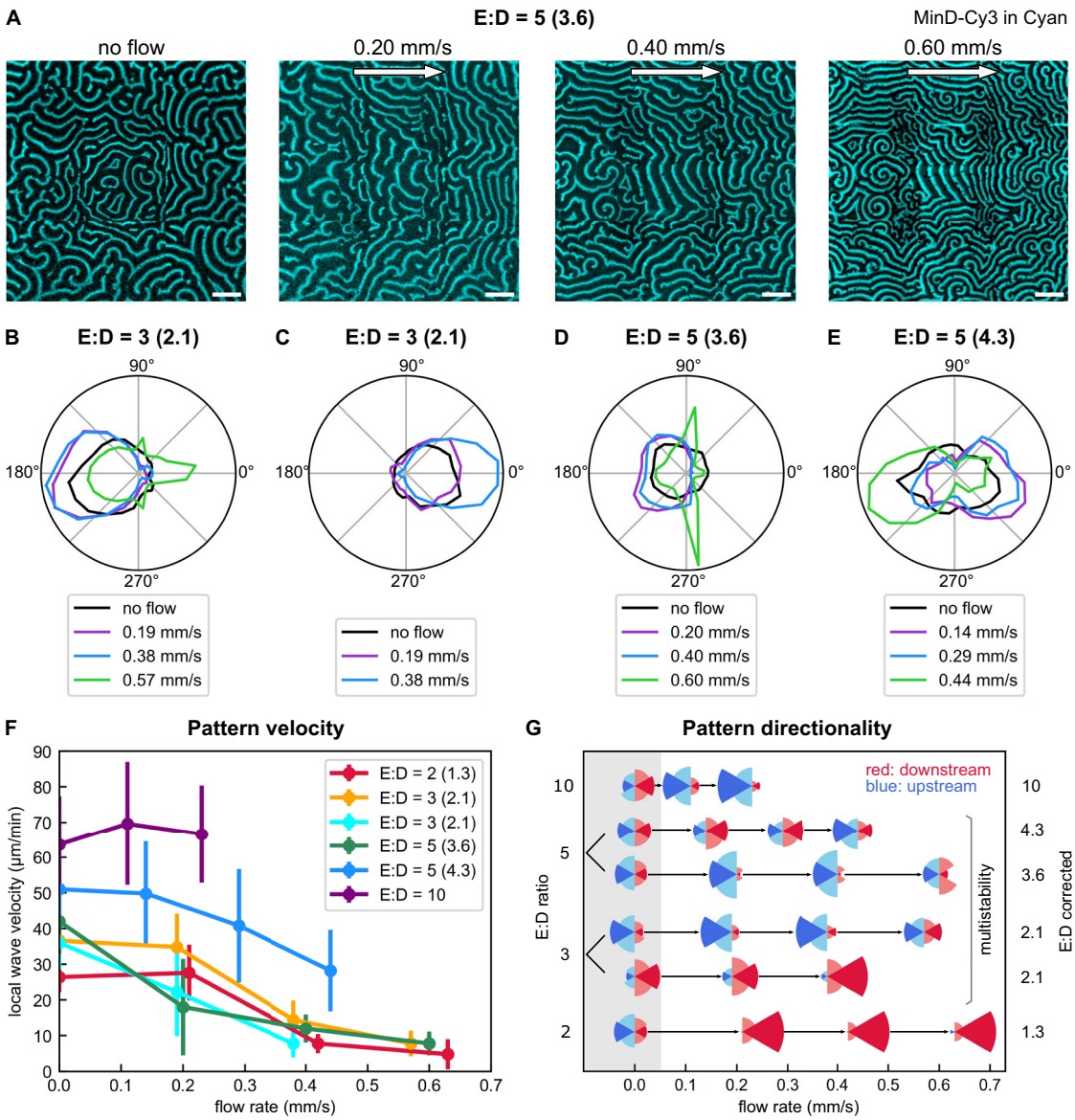

**Fig. 4 | Experimental data showing how Min patterns respond to a sequence of flow rates. A** Min patterns for different flow velocities at initial ratio E:D = 5 (corrected 3.6) sampled at the same location within the flow channel. Channel MinD-Cy3 in cyan, scale bars 100 μm. Images stitched from 3 × 3 fields of view. **B** Polar histogram showing counts per angular segment for initial ratio E:D = 3 (corrected 2.1). **C** Idem for initial ratio E:D = 3 (corrected 2.1). **D** Idem for initial ratio E:D = 5 (corrected 3.6). **E** Idem for initial ratio E:D = 5 (corrected 4.3). **F** Peak velocity magnitude ± FWHM/2 (calculated from histogram with binning size 10 nm/s) shown as function of flow rate for different E:D ratios. **G** Overview of experiments done for wildtype MinE. Downstream and upstream fractions are obtained from histogram counts per angular segment, with one series (horizontal line) representing one experiment. Size of fraction is given by the segment's radius. Red indicates downstream flowing patterns, blue indicates upstream patterns. Source data are provided as a Source Data file. Full experimental analysis is shown in the SI.

propagation (low E:D) is qualitatively captured by a parsimonious model that explicitly accounts for conformational switching of MinE. For intermediate E:D ratios, our model predicts multistability of waves propagating in either direction, resulting in hysteresis. This imparts a strong dependence of the dynamics on the initial state, which we indeed observed in our experiments for intermediate E:D ratios (Fig. 4G). Our analysis shows that different pattern-forming mechanisms operate in low and high E:D conditions, with the role of MinE bulk gradients being the key difference between these mechanisms. For large E:D ratios, we found MinE bulk gradients along the membrane to be negligible. In this way, we were able to reduce the full model to a simplified, effective model for MinD dynamics that allowed us to understand the mechanism of upstream propagation. In particular, we have shown that upstream propagation is not caused by MinE advection, but by a difference in MinD bulk gradients which promote

attachment on the upstream side. At low E:D ratios, MinE conformational switching can be neglected, as we showed both by numerical simulations and experiments using a non-switching MinE mutant. Although much has been elucidated about the role of bulk flow on pattern formation, it remains an open question at this point how downstream propagation emerges at low E:D ratios.

The key feature of our studied system is *differential* flow, i.e., the advection of different components (protein concentrations) with different velocities[41]. Bulk flow leads to advection of the proteins in solution, but not those on the membrane where the observed pattern forms. Therefore, bulk flow affects the patterns only indirectly, through the bulk-surface coupling[18] related to the attachment and detachment of proteins at the membrane surface. As a result of this indirect coupling between the protein pattern and the hydrodynamic flow, the patterns can propagate both

upstream and downstream relative to the flow direction, depending sensitively on various molecular aspects of the reaction kinetics such as the attachment–detachment dynamics. The patterns' response to flow can therefore be used as a robust, qualitative observable that allows one to identify regimes where different pattern-forming mechanisms operate. Bulk-surface coupling is a general feature of protein-based pattern formation[1]. Differential flow will generally occur when bulk flows (e.g. cytoplasmic streaming) are present in such systems[45]. On a larger scale, bulk-surface coupling is important for morphogenesis, where epithelial sheets surround fluid-filled lumens[46,47]. Signaling molecules released to the lumen will be subjected to advective flows therein. By contrast, molecules that diffuse directly from cell to cell, e.g., through gap junctions, are not subject to such flows. Here, secretion and receptor-binding are the analogs to detachment and attachment in the Min system.

From a broader perspective, advective flow is a perturbation that breaks a symmetry of the system by imposing a preferred spatial direction. Pattern formation is innately connected to symmetry breaking, to the point where the terms are sometimes even used interchangeably[48,49]. In the absence of spatial cues, symmetry breaking happens spontaneously due to the amplification of small random fluctuations or small heterogeneities within the system. As exemplified by Min patterns in the absence of flow or other cues, the propagation direction of waves is random and there is no predominant direction on average. Advective bulk flow breaks this symmetry, causing the wave patterns to align in a particular direction, either with or against the flow (as we showed). Thus, the bulk flow can be thought of as an analogue to an external magnetic field applied to a ferromagnetic material. Using such symmetry-breaking perturbations to probe these materials has provided valuable insights into the underlying physics. Here, we demonstrated that a related approach can be applied to a complex pattern-forming system that operates far from equilibrium.

Taking the experimentally observed responses to flow into account puts constraints on theoretical models. We tested two other Min models from the literature (developed by Bonny et al.[50] and Loose et al.[51]) for their propensity to produce both upstream and downstream propagating patterns (see SI, Fig. S11). We found that for the previously published kinetic rates, these models produced only downstream propagating patterns over the entire range of E:D ratios where patterns occur, even upon expanding them to include MinE-conformational switching. Thus, while our model predicted a transition between upstream and downstream propagation in qualitative agreement with experiments, we found that the other tested Min models did not. Although our simulations and in vitro experiments yielded a very similar qualitative behavior of the influence of advective bulk flow on membrane-bound Min protein patterns, shortcomings of our own model became apparent upon making quantitative comparisons to experiments. For instance, we were unable to quantitatively predict the critical E:D ratio at which the transition from downstream to upstream propagation occurs. Moreover, the dependence of wavelength and wave speed on flow speed disagree between model and experiment (compare Fig. 1E vs Fig. S19, and Fig. S9 vs Fig. S18). However, note that for the experiments, we indicate the average flow rate across the channel. The flow rate in the vicinity of the membrane (within the gradient penetration depths of a few micrometers) is significantly slower due to the Poiseuille flow profile in the channel.

The influence of advective flow on biological pattern formation has rarely been studied, both experimentally and theoretically, and remains an active topic of research[23,42,44,52–54]. Our findings on the Min protein system's response to flow suggest that additional molecular features of the protein reaction network, not yet accounted for by the current Min models, are necessary to quantitatively explain the observed phenomena. Identifying these still unknown features and including them in theoretical models remains an open issue and active topic of research. Insight into the detailed biochemical mechanisms of the MinD-MinE interactions (such as cooperative MinD self-recruitment, dimerization of MinD and MinE, MinE membrane binding, etc.) is likely needed to make progress and to allow for an accurate quantitative fit of the spatiotemporal dynamics of the Min system.

To conclude, the application of hydrodynamic flow exposed the limitations of the current models and yielded additional data that can be used to constrain models in future studies. Microfluidic applications may take advantage of Min patterns where bulk flow can be used to orient membrane-bound protein patterns and adjusting the E:D ratio allows one to decide whether one wants the protein waves to go with or against the flow. Combined with the Min system's capacity for cargo transport[31], this could offer a platform for directed transport of other membrane-associated proteins.

## Methods
Please find theoretical methods as well as extended experimental methods in the SI.

### Mathematical model
We adopt a previously established model for the Min-protein dynamics based in mass-action-law kinetics on a membrane and the adjacent bulk solution. Below, we describe the model in the simplified two-dimensional setting where the vertical dimension of the bulk has been integrated out (see SI Sec. 1.2). This model describes the concentrations of the following conformational states of MinD and MinE: Membrane-bound MinD ($m_d$); membrane-bound MinDE complex ($m_{de}$); cytosolic MinD-ADP ($c_{DD}$); cytosolic MinD-ATP ($c_{DT}$); cytosolic reactive (switch open) MinE ($c_{Er}$); cytosolic inactive (switch closed) MinE ($c_{Ei}$). The dynamics of the membrane-bound components are governed by

$$\partial_t m_d(x,t) = D_m \nabla^2 m_d + f_d, \tag{1}$$

$$\partial_t m_{de}(x,t) = D_m \nabla^2 m_{de} + f_{de}. \tag{2}$$

And the dynamics in the bulk, including advective flow, read

$$\partial_t c_{DD}(x,t) + v_f \nabla c_{DD} = D_c \nabla^2 c_{DD} + f_D - \lambda c_{DD} \tag{3}$$

$$\partial_t c_{DT}(x,t) + v_f \nabla c_{DT} = D_c \nabla^2 c_{DT} + f_{DT} + \lambda c_{DD} \tag{4}$$

$$\partial_t c_{Er}(x,t) + v_f \nabla c_{Er} = D_c \nabla^2 c_{Er} + f_{Er} - \mu c_{Er} \tag{5}$$

$$\partial_t c_{Ei}(x,t) + v_f \nabla c_{Ei} = D_c \nabla^2 c_{Ei} + f_{Ei} + \mu c_{Er} \tag{6}$$

The reaction terms $f_i$ describing the attachment, detachment and interconversion of species at the membrane as illustrated in Fig. 1C are given by

$$f_d = (k_D + k_{dD} m_d) c_D - k_{dEr} m_d c_{Er} - k_{dEi} m_d c_{Ei} \tag{7}$$

$$f_{de} = k_{dEr} m_d c_{Er} + k_{dEi} m_d c_{Ei} - k_{de} m_{de} \tag{8}$$

$$f_{DD} = k_{de} m_{de} \tag{9}$$

$$f_{DT} = -(k_D + k_{dD} m_d) c_{DT} \tag{10}$$

$$f_{Er} = -k_{dEr} m_d c_{Er} + k_{de} m_{de} \tag{11}$$

$$f_{Ei} = -k_{dEi}m_d c_{Ei}. \tag{12}$$

The above dynamics conserve the total density of MinD and MinE

$$\bar{n}_D = \int dx^2 m_d + m_{de} + c_{DD} + c_{DT}, \tag{13}$$

$$\bar{n}_E = \int dx^2 m_{de} + c_{Er} + c_{Ei}. \tag{14}$$

Together with the diffusion constants, the flow velocity and the kinetic rates, these densities are important control parameters of the dynamics. The parameter values used throughout this study are adapted from Denk et al.[22] and summarized in Table S1 in the SI. The derivation of the reduced switch model, corresponding to the interaction network cartoon in Fig. 1F, is described in SI Sec. 1.4. Parameters and equations for the full model, skeleton model and reduced switch model are given in SI Sec. 1.3. Further, we provide details on the hysteresis sweep (SI Sec. 1.5), simulations with disabled MinE advection (SI Sec. 1.6), an analysis of wave speed versus flow speed (SI Sec. 1.7) linear stability analysis (SI Sec. 1.8) as well as results obtained using other Min models (SI Sec. 1.9).

## Sample preparation

Chemicals were bought from Sigma Aldrich unless specified otherwise. Flow cells were assembled using cover and microscope slides cleaned by sonication and acid Piranha, using Parafilm as a spacer, and sealed by melting the Parafilm on a hotplate. Flow channels were about 25 mm long, 3 mm wide and 200 μm high, i.e. well above the threshold of tens of micrometers, below which coupling between upper and lower surfaces were observed[18]. Exact channel heights and widths for individual experiments are given in Tables S3 and S4. Sample channels were coated with lipid bilayers composed of DOPC:DOPG in a molar ratio 67:33 substituted with 0.01–0.02% mol TopFluor Cardiolipin, where the latter allowed us to confirm full bilayer formation before the experiment. SUVs for lipid bilayer formation were prepared via swelling followed by stepwise extrusion with final pore size of 40 nm. Before the experiment, chambers were filled with SUV solution, incubated for 1 h at 37 °C, then rinsed thoroughly with Min buffer (150 mM KCl, 25 mM TRIS pH 7.45, 5 mM MgCl$_2$). Next, the chamber (as well as the tubing for closed-circle experiments) were filled with Min protein solution comprising 1 μM MinD and MinE at a concentration in μM equal to that of the E:D ratio indicated (both including labeled fraction). The protein solution was supplemented with 2.5 mM ATP (Thermo Fisher) as well as 5 mM Phosphoenolpyruvic acid (Alfa Aesar) and 0.01 mg/mL pyruvate kinase for ATP regeneration. In order to study a wide range of flow rates, experiments with wildtype MinE were done in closed-circle systems (Fig. S12). Here, the tubing was inserted into the pump, filled with the protein solution and then connected to the pre-filled flow channel via two ports at its end. This allowed repeated recycling of the protein within the closed system. To check whether protein concentrations changed due to sticking to the comparably large internal surfaces of the tubing, we collected a portion of the original solution as well as the sample extracted from the closed-circle system at the end of every experiment and ran them side-by-side on a gel (Fig. S14) to calculate an estimate for the loss, which typically was ~30% for MinD and ~50% for MinE. In the experiments described, the corrected ratios are given along with the initial ratios. Experiments with MinE-L3E/I24N were performed with an open system, with the channel's outlet connected to the pump via tubing and its inlet connected to a reservoir of protein solution, in order to minimize loss of protein due to sticking. Laminar hydrodynamic flow was created using a pressure-driven pump (Ismatec, model IPC).

## Image acquisition

Images were acquired using an Olympus IX-81 inverted microscope equipped with an Andor Revolution XD spinning disk system and a 20x objective (Olympus PlanN 20x / 0.4 NA). For excitation of MinD-Cy3 and MinE-Cy5, laser lines 561 nm and 640 nm were used. Images were acquired at multiple positions in grid geometries with some overlap between adjacent fields of view (as shown in Fig. S12) at 15 s intervals (4 frames per minute). Up to three comparably sized regions per sample were imaged (example given in Fig. S20), obtaining total covered areas of up to 2 mm². Regions were located in the central third of the channel to avoid possible turbulence close to the inlet and outlet. Following 1 h of incubation at room temperature, images were first recorded without flow at the selected positions in order to get a reference for comparing subsequential acquisitions to. The flow rate was then set and increased incrementally as indicated. The average cross-sectional flow rate (in mm/s) was calculated from the flow channel's width and height as well as the pump's set flow rate (in volume/time). Images at the chosen flow rates and positions were recorded following at least 15 min of incubation, as we empirically found this to be about the time the pattern needed to respond to the new flow rate.

## Image analysis

Image cleaning, stitching as well as wave crest detection and propagation analysis were done using custom-built MatLab and Python scripts using the methodology outlined in Meindlhumer et al.[43]. More details are also given in the SI. In brief, we identified wave crests in each frame within a stack using phase images, then compared sequential images to obtain the translation of each crestpoint frame to frame. The result of our analysis is a collective list of vectors ($v_x$, $v_y$) found at positions ($x$, $y$) of individual frames, collected from up to three comparable imaging regions. These vectors were analyzed with respect to their magnitude and directionality. Very low propagation velocities were cropped to eliminate the influence of static objects (protein aggregates) that would otherwise show in the results. Velocity vectors were excluded if they had a magnitude below 10% of the median. Analysis shown in the paper was performed on the patterns acquired for MinD-Cy3 unless noted otherwise. The full pattern direction analysis results for all experiments performed can be found in the SI.

## Data availability

The simulation and experimental data created for this study are freely available on Zenodo at https://doi.org/10.5281/zenodo. 7339803[55]. Source data are provided with this paper.

## Code availability

The code used in this study is available upon request to the corresponding author.

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

## Acknowledgements

We thank Jacob Halatek for insightful discussions and initial, preliminary work on this project. We thank Petra Schwille at the MPI of Biochemistry Martinsried for kindly providing plasmids for overexpression of MinE-L3E/I24N. Further, we thank Jaco van der Torre for wet lab support, Jérémie Capoulade for support with spinning disc confocal microscopy and finally, Eli van der Sluis and Ashmiani van den Berg for purification of MinE-L3E/I24N. C.D. acknowledges the support provided by the BaSyC —"Building a Synthetic Cell" Gravitation grant (024.003.019) of the Netherlands Ministry of Education, Culture and Science (OCW) and the Netherlands Organisation for Scientific Research (NWO). E.F. acknowledges support from the German Research Foundation DFG through Collaborative Research Center SFB 1032, Project- ID No. 201269156. E.F. acknowledges support from Germany's Excellence Strategy, Excellence Cluster ORIGINS, EXC-2094-390783311. J.F. acknowledges the Ad Futura Scholarship (244. javni razpis) from the Public Scholarship, Development, Disability and Maintenance Fund of the Republic of Slovenia.

## Author contributions

E.F., C.D., and F.B. designed research; S.M. carried out experiments; J.K. and S.M. performed microscope image data analysis; E.F., F.B., and J.F. designed the theoretical models; F.B. and J.F. performed the mathematical analyses and simulations; E.F. supervised the theoretical work; C.D. supervised the experimental work; S.M., F.B., and J.F. wrote the paper, with input from all authors.

## Funding

## Competing interests

The authors declare no competing interests.
