## [Peer Review File · Nature Communications]

Directing Min protein patterns with advective bulk flowREVIEWER COMMENTS

Reviewer #1 (Remarks to the Author):

In 2014, Vecchiarelli et al. noted a peculiar phenomenon when reconstituting Min patterning de novo in flow chambers – MinDE waves propagated upstream. It was hypothesized that upstream propagation was due to activated or “lingering” MinE molecules rapidly rebinding the bilayer downstream of the initial binding zone. The resulting asymmetry in active MinE molecules asymmetrically remove MinD from the bilayer. While the downstream-facing half of the MinD zone is removed from the bilayer, the upstream-facing half is left unhindered to expand into a propagating wave that travels upstream. This was simply a hypothesis and not directly tested. Here the authors provide evidence towards an alternative model whereby the upstream wave propagation is likely not due to downstream transport of reactive MinE. Instead, the authors provide convincing in silico and cell-free reconstitution evidence implicating advective bulk flow as the driver of pattern directionality. The results are certainly noteworthy.

The work will be of significance to the fields of prokaryotic cell biology and synthetic biology. Although the Min system has been reconstituted in a variety of ways, the use of flow to probe pattern formation and underlying mechanisms is original and the data is superb.

The conclusions and claims in the paper are well justified for the most part, and shortcomings of the in silico models are accounted for. As with the vast majority of Min modelling to date, the authors use MinD auto-cooperativity as the mode of self-recruitment to the membrane. The experimental evidence for MinD cooperatively binding membrane continues to be very weak/non-existent, while direct evidence implicating MinE in stimulating MinD recruitment to the bilayer is relatively strong, but continues to be ignored. I understand why models continue to use auto-cooperative MinD membrane binding – its simplicity. I’m not too concerned about this issue here because this parameter likely does not change the main findings and conclusions of this paper – advective bulk flow can direct Min patterning. Therefore, I don’t believe additional evidence is needed for publication.

The methodology is sound, the work exceeds current standards in the field, and there were no notable flaws in the data analysis, interpretation and conclusions that would prohibit publication or require revision.

Reviewer #2 (Remarks to the Author):

This manuscript of Meindlhumer et al. reports that the direction of the propagating Min protein patterns can be altered by the bulk flow of solution both experimentally and theoretically. The numerical models are developed based on the previous model from the authors’ groups that is further distinguished into the reduced switch model and the skeletal model by the E:D ratio. This is based on an assumption that abundant MinE may mask the contribution of latent MinE. The two models correspond to different wave features occurring upstream and downstream of the flow. New parameters and values are introduced into the numerical models to simulate effect of the bulk flow on the surface wave, revealing that the wave can propagate in a reversed direction in responding to the flow rate. The significance of using the bulk flow to control the wave direction appears to be more specialized for physical or engineering purposes than for understanding the molecular basis of the pattern formation.

Page 3:

1. Not sure about the claim of using the bulk flow to probe molecular mechanism, since the bulk flow is external to the functioning Min system. Meanwhile, the factor of the E:D

ratio is well characterized in previous works by the laboratories of Mizuuchi and Schwille.

2. Not sure about the analogies to actomyosin cortical flow that the is far complicated than the Min system at the molecular level, and to cytosolic streaming that normally refers to organelles and vacuoles that are transporting along cytoskeletal filaments.

3. " Min protein surface patterns tend to align in wave fronts perpendicular to the direction of flow, and that observables such as their preferred direction of propagation can be linked to underlying molecular mechanisms." - The description is abstractive.

Page 4, Fig 1D,E, Fig. 2:

4. In previous works, for example Denk et al. 2018, the skeletal model involving the latent MinE was used to model the propagating Min protein waves with a wide range of the E:D ratios. The question here is why additional reduced switch model is necessary to generate the upstream pattern which looks similar to most observations in previous works, while the skeletal model generates the downstream propagation pattern that differs from the previous works?

5. In Fig 2, is it the case that the left micrograph shows the upstream wave and the right shows the downstream wave? They look similar and not consistent with the simulation in Fig 1E. Does the pattern difference between traveling and spiral waves count? or just the direction?

6. The way that the authors measure the wave direction and transform into a polarity plot is hard to understand.

Pages 5, 6.

7. It seems that the author intends to explain the direction and pattern changes may be caused by the difference of E:D ratio in the bulk solutions of MinD and MinE undergoing advection. However, it is well studied that modulating the E:D ratio leads to changes in the wave patterns. It is not convincing to think that this molecular mechanism could be altered by apply advection of the protein solution on top of the system.

8. Two variables, flow rate of the bulk solution and E:D ratio, complicated the interpretation. The cause-and-result relation is arguable.

9. Does "hysteresis" refer specifically to the transition point or the same as bistable (multistable) phase? Is it the black dash line in Fig 3A?

Pages 6, 7.

10. In Fig 3A, the downstream waves occur below E:D ratio of 0.4 at all flow rates. This is below all ratios (wild-type MinE) used in the simulation and the supporting experiments.

11. In Fig 4A, are micrographs sampled in the same image field but with increasing speed?

Reviewer #3 (Remarks to the Author):

In this work, the authors studied the effects of hydrodynamic flow on the pattern formation of the Min system. The Min system is the main playground of reaction-diffusions equations in intracellular systems. The system has relatively simpler ingredients among complex biological phenomena, but still, it shows interesting patterns. The molecular mechanism is relatively well understood, if not complete, by using theoretical models with reaction-diffusion equations. The authors performed numerical simulations of the model with the advection term in bulk (cytosol), and investigated whether generated waves are propagating downstream or upstream. The theoretical analyses are compared with experimental observations. The effect of flow has already been observed in Ref.[23]. The novelty in this work is (I) clarification of the mechanism using the theoretical model in comparison to the experiments, and (II) detailed analyses of the dependence of the flow effects on MinE/MinD concentration ratios and flow velocity, such as (II-A) direction of wave propagation (upstream, downstream, and coexistence), (II-B) wave speed, and (II-C) wavelength. In terms of

the mechanism, after performing the simulations of the full model, the authors proposed the two reduced models that dominantly reproduce (I-A) upstream and (I-B) downstream wave propagation, respectively. In particular, the authors found that the advection of the depleted zone of MinD in bulk is a key mechanism of upstream wave propagation. This is the challenge to the hypothesis in Ref.[23].

The authors show both theoretically and experimentally upstream wave propagation under a high MinE/MinD ratio, and downstream propagation for the opposite case (II-A). However, the authors failed to explain the wave width (II-C), and the mechanism of the downstream wave propagation. I could not find the dependence of the wave speed on flow velocity and MinE/MinD ratio in the theoretical model, but only for the experimental results. Even for (II-A), the agreement between simulations and experiments is qualitative, not quantitative. The downstream wave propagation occurs through flow-induced instability in the model, but this is not the case in the experiments. In addition, the critical MinE/MinD ratio seems to be very different. I should stress that, on the one hand, the model is complex (six-variable nonlinear PDEs) and, therefore, is too difficult to understand everything. On the other hand, the model is too simple to describe every aspect of the experimental results. My intention here is to summarise the current situation in this work.

Whether this study has a broad readership or not is a delicate issue. Indeed, the results are valid only for the specific Min system, and biological relevance is not clear as the flow does not seem to be important for cell division of E-coli. Still, I personally think it does have a broad readership, because the pattern formation with mass conservation and bulk-surface coupling is shared by other systems, in which flow may be important. The finding in this work could be applied to other systems.

Overall, I think this is nice work. The idea of differential flow is applied to the Min system, and indeed, the wave propagation is dependent on the flow. As I discussed, it may have a broad impact on the pattern formation of intracellular systems. Nevertheless, I have a big concern that the claims are not solidly justified by the data in the current manuscript. Therefore, I hesitate to recommend this work for publication in Nature Communications. Hereafter, I shall discuss the issues on each topic.

(I) Mechanism

(i) The authors mentioned that "reduced switch model, ... exclusively exhibits upstream propagation in response to flow" and "reduced skeleton model ..., we exclusively found downstream propagating waves," but no data was shown even in Supplementary Information, as far as I understood. I think it is necessary to show the phase diagram similar to Fig.3A.

(ii) To derive the reduced switch model, the authors assumed not only high MinE/MinD ratio, but also other conditions, small k_{dEi} , large μ , and QSSA (I guess this means large v_f , D_c , and μ). It is not clear to me how these conditions are met.

(iii) I am curious why the flow velocity is the order of 0.1mm/s, which is much faster than the wave speed $\sim 1\mu\text{m/s}$. This means that the advection in bulk is much faster than the wave propagation on the membrane. I would imagine that the advection is balanced with the diffusion, and therefore, I would expect the flow velocity such as $v \sim (D/\text{time scale})^{1/2}$. The time scale of the chemical reactions is $\sim \text{sec}$, and then, $v \sim 1\text{-}10\mu\text{m/s}$, which looks much slower than the actual flow velocity. My concern is that if the advection is much faster than the wave propagation, I am not sure the mechanism of Fig.1B can apply. The fast advection may lead to smoothing out the MinD gradient.

(iv) In Discussion, the authors claim that other models produced exclusively downstream propagating patterns, again without any data. In Sec.1.8 in SI, the authors merely explained each model and its simulation condition, but no data was shown. I do not think it is convincing to claim one model is superior to others without comparing actual data. I should also point out that such comparison is delicate because there are

many parameters in those models, which we cannot specify. It is possible that by changing the parameters, other models can reproduce the same behaviours found in this study. I think we should be very careful about the issue.

(II-A) Wave direction

(v) In experiments, I think the change in the direction of wave propagation by the flow direction is clear. I tend to believe the results of the MinE/MinD dependence. However, among the four ratios (before correction), the two intermediate ratios show multistability, and only one condition is demonstrated each for upstream/downstream propagation. I think it is more convincing to add a few more data for upstream-dominated and downstream-dominated cases.

(vi) It is not clear to me why the L3I24N mutant at the low MinE/MinD ratio justifies the reduced skeleton model. If the L3I24N mutant eliminates the switch between reactive and latent states of MinE, shouldn't it reproduce downstream propagation at the high MinE/MinD ratio? This is not clear partially because the data of the skeleton model under flow is not shown.

(II-B) Wave speed

(vii) Is the wave speed in the theoretical model slower at higher flow velocity? I think the speed decreases in experiments, and the tendency looks clearer.

(II-C) Wavelength

(viii) As far as I had a look at the movies, the width of the peaks of waves looks wider for higher flow velocity. I may be too naïve, but is it possible that wave width is, in fact, wider by using another way to measure the width? If the spectrum near the unstable wavenumber is flatter, many modes are excited under higher flow velocity.

(ix) I personally think that no dependence of wavelength on the flow velocity in experiments is a bit serious for understanding the mechanism of downstream wave propagation. This is because, according to the linear stability analyses, the downstream wave propagation appears as a second peak in the spectrum near $k \sim 0$. If the wavelength of the downstream propagation is not different from that of upstream propagation, a different mechanism may be happening for the downstream propagation. I think it is better to clarify this issue.

DETAILED COMMENTS:

(1) Fig.4

I think the figure captions of B-E do not fit the titles of their figures.

(2) Sec.1.2 and Fig.S2 in Supplementary Information

I think the justification for using reduced bulk is made only for downstream wave propagation under a low MinE/MinD ratio and upstream propagation under a high ratio. The behaviours near the coexistence region and dependence on the flow velocity are not guaranteed. I am also puzzled by the range of the height in Fig.S2. I guess the height in the experiments is 180um-230um, whereas it is 1-30um. Why did the authors choose those heights?

REPLY TO THE REVIEWER COMMENTS

Reviewer #1:

In 2014, Vecchiarelli et al. noted a peculiar phenomenon when reconstituting Min patterning de novo in flow chambers – MinDE waves propagated upstream. It was hypothesized that upstream propagation was due to activated or “lingering” MinE molecules rapidly rebinding the bilayer downstream of the initial binding zone. The resulting asymmetry in active MinE molecules asymmetrically remove MinD from the bilayer. While the downstream-facing half of the MinD zone is removed from the bilayer, the upstream-facing half is left unhindered to expand into a propagating wave that travels upstream. This was simply a hypothesis and not directly tested. Here the authors provide evidence towards an alternative model whereby the upstream wave propagation is likely not due to downstream transport of reactive MinE. Instead, the authors provide convincing in silico and cell-free reconstitution evidence implicating advective bulk flow as the driver of pattern directionality. The results are certainly noteworthy.

The work will be of significance to the fields of prokaryotic cell biology and synthetic biology. Although the Min system has been reconstituted in a variety of ways, the use of flow to probe pattern formation and underlying mechanisms is original and the data is superb.

The conclusions and claims in the paper are well justified for the most part, and shortcomings of the in silico models are accounted for. As with the vast majority of Min modelling to date, the authors use MinD auto-cooperativity as the mode of self-recruitment to the membrane. The experimental evidence for MinD cooperatively binding membrane continues to be very weak/non-existent, while direct evidence implicating MinE in stimulating MinD recruitment to the bilayer is relatively strong, but continues to be ignored. I understand why models continue to use auto-cooperative MinD membrane binding – its simplicity. I'm not too concerned about this issue here because this parameter likely does not change the main findings and conclusions of this paper – advective bulk flow can direct Min patterning. Therefore, I don't believe additional evidence is needed for publication.

The methodology is sound, the work exceeds current standards in the field, and there were no notable flaws in the data analysis, interpretation and conclusions that would prohibit publication or require revision.

We sincerely thank the reviewer for their kind and insightful feedback. We were particularly glad to hear they valued our data analysis and overall agreed with our choice of models. As they also point out, an important conclusion of our paper is that the currently available models seem to be missing out on some property of this system (likely related to cooperativity), and that future research is needed to clarify these aspects.

Reviewer #2:

This manuscript of Meindlhumer et al. reports that the direction of the propagating Min protein patterns can be altered by the bulk flow of solution both experimentally and theoretically. The numerical models are developed based on the previous model from the authors' groups that is further distinguished into the reduced switch model and the skeletal model by the E:D ratio. This is based on an assumption that abundant MinE may mask the contribution of latent MinE. The two models correspond to different wave features occurring upstream and downstream of the flow. New parameters and values are introduced into the numerical models to simulate effect of the bulk flow on the surface wave, revealing that the wave can propagate in a reversed direction in responding to the flow rate. The significance of using the bulk flow to control the wave direction appears to be more specialized for physical or engineering purposes than for understanding the molecular basis of the pattern formation.

Page 3:

1. Not sure about the claim of using the bulk flow to probe molecular mechanism, since the bulk flow is external to the functioning Min system. Meanwhile, the factor of the E:D ratio is well characterized in previous works by the laboratories of Mizuuchi and Schwille.

We thank the reviewer for pointing out this aspect. We agree that the link between bulk flow (a macroscopic perturbation) and molecular mechanisms (which operate on a microscopic level) is not clear a priori - and hence it is exciting that we see dramatic effects of the bulk flow on the surface patterns.

In particular, we show that microscopic details affect *how* the patterns respond to advective bulk flow. Therefore, the bulk flow serves as an *indirect* probe for the microscopic details. Here, mathematical models play a key role by linking the microscopic level to the macroscopic observations, similar as in our previous theoretical work on the role of the MinE switch on pattern robustness (Ref. [22], Denk et al, 2018).

We have revised the introduction to emphasise that the bulk flow can serve as an indirect probe for molecular details and highlight the role of mathematical models.

2. Not sure about the analogies to actomyosin cortical flow that the is far complicated than the Min system at the molecular level, and to cytosolic streaming that normally refers to organelles and vacuoles that are transporting along cytoskeletal filaments.

Indeed, the molecular machineries giving rise to flows in biological systems are complex and often involve regulatory feedback loops. However, here we simply wanted to make the point that there are cases of advective transport of molecules in biological systems. This is a generic property of flows, irrespective of their origin and regulation.

To avoid confusion, we have slightly rephrased the respective passage.

3. "Min protein surface patterns tend to align in wave fronts perpendicular to the direction of flow, and that observables such as their preferred direction of propagation can be linked to underlying molecular mechanisms." - The description is abstractive.

Aiming for better clarity, we slightly rephrased the cited passage.

Page 4, Fig 1D,E, Fig. 2:

4. In previous works, for example Denk et al. 2018, the skeletal model involving the latent MinE was used to model the propagating Min protein waves with a wide range of the E:D ratios. The question here is why additional reduced switch model is necessary to generate the upstream pattern which looks similar to most observations in previous works, while the skeletal model generates the downstream propagation pattern that differs from the previous works?

The simulations shown in Fig. 1D,E and the phase diagram in Fig. 3 were obtained with the full switch model that was previously established in Denk et al. 2018 [22]. This model exhibits both upstream propagating patterns for high E:D ratios and downstream propagating patterns for low E:D ratios.

We have revised the caption of Fig. 1 to clarify this point.

The reduced switch model is not necessary to generate upstream propagating patterns. Instead, its purpose is to help us understand the self-organisation mechanism that underlies them. This leads to the qualitative description shown in Fig. 1 B for the case of high E:D ratio.

5. In Fig 2, is it the case that the left micrograph shows the upstream wave and the right shows the downstream wave? They look similar and not consistent with the simulation in Fig 1E. Does the pattern difference between traveling and spiral waves count? or just the direction?

In Fig. 2, we show the “no flow” (left half) and “flow” (right half) cases for a high E:D ratio (10), a low E:D ratio (2) and the MinE mutant.

Indeed, there appear to be differences in the detailed pattern morphology, and a deeper analysis thereof might be an interesting and worthwhile study in itself. However, within the scope of this paper, there is no need for a detailed description of these morphological tendencies. We chose to focus on the patterns' propagation direction, as this was the most accessible observable, both in simulations and experiments.

6. The way that the authors measure the wave direction and transform into a polarity plot is hard to understand.

We thank the reviewer for pointing out that they found it difficult to follow our representation of local analysis results.

To make up for the shortcomings in our verbal description, we now tried to partly rephrase it in the text. Importantly, we added a new figure to the SI (Fig. S13), which is meant to tie up loose ends and connect the different types of local analysis/plots for experimental data used throughout the paper and SI.

Pages 5, 6.

7. It seems that the author intends to explain the direction and pattern changes may be caused by the difference of E:D ratio in the bulk solutions of MinD and MinE undergoing advection. However, it is well studied that modulating the E:D ratio leads to changes in the wave patterns. It is not convincing to think that this molecular mechanism could be altered by apply advection of the protein solution on top of the system.

The molecular reaction-diffusion processes that underlie Min-protein patterns contain two main elements: On the one hand, there are molecular interactions (chemical reactions), on the other hand, there is spatial transport of the molecules by diffusion and advection. On the macroscopic level, patterns emerge from the collective interplay between these two processes.

We do not claim that advective flow changes the individual molecular interactions. Rather, advective flow affects the transport of molecules and therefore has an influence on the collective level. Details of the molecular interactions, and the ratio of protein concentrations, can therefore influence the response of a macroscopic pattern to advective flow (i.e. whether the pattern propagates upstream or downstream).

8. Two variables, flow rate of the bulk solution and E:D ratio, complicated the interpretation. The cause-and-result relation is arguable.

We show that the flow rate and the E:D ratio are both important parameters. The structure of the phase diagram and the observation of hysteresis/multistability show that effects of E:D ratio and flow rate are interconnected such that there is no one-to-one cause-effect relationship. As a baseline control, we always included the no-flow case.

9. Does “hysteresis” refer specifically to the transition point or the same as bistable (multistable) phase? Is it the black dash line in Fig 3A?

Hysteresis does not refer to the transition point but to the phenomenon that the position of the transition point depends on the direction of the transition. These transitions are indicated by the blue and red line in Fig. 3A, respectively. They enclose a region of bi-/multistability, where multiple types of steady-state patterns can be observed depending on the initial condition. The grey dashed line indicates the transition from downstream to upstream propagation for homogeneous initial conditions with small random noise added.

Pages 6, 7.

10. In Fig 3A, the downstream waves occur below E:D ratio of 0.4 at all flow rates. This is below all ratios (wild-type MinE) used in the simulation and the supporting experiments.

The reviewer observes correctly that the experiment and theory do not agree on a quantitative level. We explicitly point out these discrepancies in the text and discuss that the current model likely misses molecular details that would be needed to reach quantitative agreement. In fact, we are currently exploring various possible model extensions but the results are too preliminary yet to discuss.

11. In Fig 4A, are micrographs sampled in the same image field but with increasing speed?

Correct. We added an extra note in the text and figure description to clarify this point.

Reviewer #3:

In this work, the authors studied the effects of hydrodynamic flow on the pattern formation of the Min system. The Min system is the main playground of reaction-diffusions equations in intracellular systems. The system has relatively simpler ingredients among complex biological phenomena, but still, it shows interesting patterns. The molecular mechanism is relatively well understood, if not complete, by using theoretical models with reaction-diffusion equations. The authors performed numerical simulations of the model with the advection term in bulk (cytosol), and investigated whether generated waves are propagating downstream or upstream. The theoretical analyses are compared with experimental observations. The effect of flow has already been observed in Ref.[23]. The novelty in this work is (I) clarification of the mechanism using the theoretical model in comparison to the experiments, and (II) detailed analyses of the dependence of the flow effects on MinE/MinD concentration ratios and flow velocity, such as (II-A) direction of wave propagation (upstream, downstream, and coexistence), (II-B) wave speed, and (II-C) wavelength. In terms of the mechanism, after performing the simulations of the full model, the authors proposed the two reduced models that dominantly reproduce (I-A) upstream and (I-B) downstream wave propagation, respectively. In particular, the authors found that the advection of the depleted zone of MinD in bulk is a key mechanism of upstream wave propagation. This is the challenge to the hypothesis in Ref.[23].

We thank the reviewer for carefully reading our manuscript, which is excellently summarised above, and for providing valuable feedback.

The authors show both theoretically and experimentally upstream wave propagation under a high MinE/MinD ratio, and downstream propagation for the opposite case (II-A). However, the authors failed to explain the wave width (II-C), and the mechanism of the downstream wave propagation. I could not find the dependence of the wave speed on flow velocity and MinE/MinD ratio in the theoretical model, but only for the experimental results.

Indeed, we do not currently have an heuristic explanation for downstream propagation. Our attempts to narrow down the relevant players have revealed that a complex interplay of the reactive dynamics with MinD advection and MinE diffusion in the bulk is responsible for this phenomenon. Unfortunately, we have not been able to further disentangle this interplay. Moreover, understanding detailed properties of highly nonlinear waves, such as the width, is notoriously difficult and goes beyond the scope of this work.

Regarding the wave speed, we have added data from numerical simulations to the SI. In the simulations, the wave speed first decreases for low flow rates and then increases for higher flow rates. In contrast, experiments only show decreasing wave speeds. This further emphasises that further model extensions will be required to reproduce the experimental observations quantitatively.

Even for (II-A), the agreement between simulations and experiments is qualitative, not quantitative. The downstream wave propagation occurs through flow-induced instability in the model, but this is not the case in the experiments. In addition, the critical MinE/MinD ratio seems to be very different. I should stress that, on the one hand, the model is complex (six-variable nonlinear PDEs) and, therefore, is too difficult to understand everything. On the

other hand, the model is too simple to describe every aspect of the experimental results. My intention here is to summarise the current situation in this work.

We agree fully with the reviewer's remarks. Indeed, we consider it to be among the important conclusions of this paper that our models as of now are not yet able to capture every aspect of this simple, yet astonishingly rich system.

To further highlight this aspect, we slightly modified parts of the text in the discussion.

Whether this study has a broad readership or not is a delicate issue. Indeed, the results are valid only for the specific Min system, and biological relevance is not clear as the flow does not seem to be important for cell division of E-coli. Still, I personally think it does have a broad readership, because the pattern formation with mass conservation and bulk-surface coupling is shared by other systems, in which flow may be important. The finding in this work could be applied to other systems.

Overall, I think this is nice work. The idea of differential flow is applied to the Min system, and indeed, the wave propagation is dependent on the flow. As I discussed, it may have a broad impact on the pattern formation of intracellular systems. Nevertheless, I have a big concern that the claims are not solidly justified by the data in the current manuscript. Therefore, I hesitate to recommend this work for publication in Nature Communications. Hereafter, I shall discuss the issues on each topic.

We thank the reviewer for this positive assessment of our work. We agree that we could have done a better job at providing a fuller set of data to support our claims. As detailed below, we have now added supplementary datasets (shown in Figs. S3, S4, S5, S28, S29, S30) to support our claims. We hope that we addressed all the reviewer's remaining concerns.

(I) Mechanism

(i) The authors mentioned that "reduced switch model, ... exclusively exhibits upstream propagation in response to flow" and "reduced skeleton model ..., we exclusively found downstream propagating waves," but no data was shown even in Supplementary Information, as far as I understood. I think it is necessary to show the phase diagram similar to Fig.3A.

We have added two figures to the SI showing that the skeleton model exclusively exhibits downstream propagation (Fig. S5) while the reduced switch model exhibits only upstream propagation (Fig. S3).

(ii) To derive the reduced switch model, the authors assumed not only high MinE/MinD ratio, but also other conditions, small k_{dEi} , large μ , and QSSA (I guess this means large v_f , D_c , and μ). It is not clear to me how these conditions are met.

As we describe in the text, the derivation of the reduced switch model requires taking the double limit of high E:D ratio and fast switching (large μ). The former limit automatically implies that k_{dEi} needs to be small, as the reviewer points out, since the product $n_E k_{dEi}$ determines the effective MinE recruitment rate (see SI). Similarly, large μ requires large k_{dEr} ,

since switching and recruitment compete for reactive MinE. Small k_{dEi} and large μ and k_{dER} are motivated by experimental estimates as discussed in Ref. [22]. Together, these assumptions allow one to make a QSSA on the local level of the reaction dynamics, i.e. disregarding diffusive and advective transport. In the spatially coupled setting, QSSA is more subtle because one needs to compare not only timescales but also length scales. For instance, the ratio of bulk diffusivity to switching rate determines the penetration depth of the reactive MinE layer into the bulk (see text above Eq. (17) in the SI). The QSSA requires that local concentrations are not driven too far away from their instantaneous, local equilibrium values by transport processes. In other words, QSSA holds for not too large v_f and D_c .

We have revised the SI to clarify these questions raised by the reviewer.

(iii) I am curious why the flow velocity is the order of 0.1mm/s, which is much faster than the wave speed $\sim 1\mu\text{m/s}$. This means that the advection in bulk is much faster than the wave propagation on the membrane. I would imagine that the advection is balanced with the diffusion, and therefore, I would expect the flow velocity such as $v \sim (D/\text{time scale})^{1/2}$. The time scale of the chemical reactions is $\sim \text{sec}$, and then, $v \sim 1\text{-}10\mu\text{m/s}$, which looks much slower than the actual flow velocity. My concern is that if the advection is much faster than the wave propagation, I am not sure the mechanism of Fig.1B can apply. The fast advection may lead to smoothing out the MinD gradient.

Indeed, for sufficiently fast flow the cytosol gradient is smoothed out such that the effect on the propagation direction becomes smaller. As a consequence, for a two-component model that produces stationary patterns in the absence of flow, the pattern propagation in the presence of flow becomes slower for fast flow (Ref. [44]).

However, for the flow speeds that we consider, there are still significant bulk gradients as shown in the new SI figure S4. This is enough to break the upstream-downstream symmetry, which then biases the wave propagation direction.

Finally, please note that we indicate the *average* flow rate across the channel. The flow rate in the vicinity of the membrane (within the gradient penetration depths of a few micrometres) is significantly slower due to the Poiseuille flow profile in the channel.

(iv) In Discussion, the authors claim that other models produced exclusively downstream propagating patterns, again without any data. In Sec.1.8 in SI, the authors merely explained each model and its simulation condition, but no data was shown. I do not think it is convincing to claim one model is superior to others without comparing actual data. I should also point out that such comparison is delicate because there are many parameters in those models, which we cannot specify. It is possible that by changing the parameters, other models can reproduce the same behaviours found in this study. I think we should be very careful about the issue.

We have added a figure showing typical kymographs for the Loose et al. model and the Bonny et al. model, both of which show downstream propagation (Fig. S11). We agree with the reviewer that it cannot be ruled out that these models possibly may exhibit downstream propagation in a different parameter regime, although we find this unlikely. Only the kinetic rates and diffusivities used in the original studies were tested. In addition, we have varied the E:D ratio across the whole range where patterns form.

We have revised the passage in the main text to clarify these details.

(II-A) Wave direction

(v) In experiments, I think the change in the direction of wave propagation by the flow direction is clear. I tend to believe the results of the MinE/MinD dependence. However, among the four ratios (before correction), the two intermediate ratios show multistability, and only one condition is demonstrated each for upstream/downstream propagation. I think it is more convincing to add a few more data for upstream-dominated and downstream-dominated cases.

We thank the reviewer for having such a thorough look at the experimental data. To support our claims, we now added a few additional datasets which show upstream/downstream propagation at high/low E:D ratios (see SI, Figs. S28 - S30).

These datasets were recorded in slightly deviating experimental conditions and for smaller imaged areas, but nonetheless show the same trend as the closed-circle experiments summarised in Fig. 4.

(vi) It is not clear to me why the L3I24N mutant at the low MinE/MinD ratio justifies the reduced skeleton model. If the L3I24N mutant eliminates the switch between reactive and latent states of MinE, shouldn't it reproduce downstream propagation at the high MinE/MinD ratio? This is not clear partially because the data of the skeleton model under flow is not shown.

In contrast to the switch model, the 'skeleton model' shows no pattern formation for high E/D ratios (see Fig. S5). This is in agreement with experimental observations for the L3E/I24N mutant, which forms patterns only for low E/D ratios (around 0.05, compare Ref. [22]). Therefore, in the high E/D regime where the full switch model and the wild-type experimental system exhibit upstream propagating waves, the skeleton model and the L3E/I24N mutant system don't show any patterns.

Side note: We apologise for the typo in the MinE mutant's nomenclature, which is now also corrected in the text.

(II-B) Wave speed

(vii) Is the wave speed in the theoretical model slower at higher flow velocity? I think the speed decreases in experiments, and the tendency looks clearer.

As we have written above, the simulations don't show a systematic decrease of wave speed with flow speed. Rather, the wave speed first decreases and then starts to increase.

(II-C) Wavelength

(viii) As far as I had a look at the movies, the width of the peaks of waves looks wider for higher flow velocity. I may be too naïve, but is it possible that wave width is, in fact, wider by using another way to measure the width? If the spectrum near the unstable wavenumber is flatter, many modes are excited under higher flow velocity.

Within the scope of this paper, we did not quantify the wave width as such. Instead, we quantified the patterns' local propagation velocities (as shown in Figs. 4, S18) as well as in global wavelength via autocorrelation analysis (Figs. S19), following the methodology outlined in Ref. [43] (also summarised in the SI).

We attempted a quantification of the wave width via full-width-at-half-maximum using our local analysis approach, but could not find a clear correlation between the wave width and the applied flow velocity. However, we must note that our method is at the moment not optimised for this purpose.

(ix) I personally think that no dependence of wavelength on the flow velocity in experiments is a bit serious for understanding the mechanism of downstream wave propagation. This is because, according to the linear stability analyses, the downstream wave propagation appears as a second peak in the spectrum near $k \sim 0$. If the wavelength of the downstream propagation is not different from that of upstream propagation, a different mechanism may be happening for the downstream propagation. I think it is better to clarify this issue.

We thank the reviewer for their insightful criticism of our models' shortcomings. Indeed, as we point out in the discussion, at the moment we lack a deeper understanding of downstream propagation and are unable to come up with a heuristic argument for its mechanism (such as given for upstream propagation in Fig. 1 B).

As also mentioned above, we slightly rephrased the discussion to hopefully highlight this aspect more clearly.

DETAILED COMMENTS:

(1) Fig.4

I think the figure captions of B-E do not fit the titles of their figures.

We thank the reviewer for noticing this and apologise for our oversight. We have now corrected the order of figure captions.

(2) Sec.1.2 and Fig.S2 in Supplementary Information

I think the justification for using reduced bulk is made only for downstream wave propagation under a low MinE/MinD ratio and upstream propagation under a high ratio. The behaviours near the coexistence region and dependence on the flow velocity are not guaranteed. I am also puzzled by the range of the height in Fig.S2. I guess the height in the experiments is 180 μm -230 μm , whereas it is 1-30 μm . Why did the authors choose those heights?

Above ca. 30 μm , changing the bulk height no longer affects patterns because diffusion effectively screens the more distant bulk from the membrane. The simulations at bulk heights below 30 μm are done to show that the propagation direction is independent of bulk height even in the regime where it affects other features of the patterns (see Ref. [18]). This allows us to eliminate the extended bulk and perform simulations in a reduced geometry where the bulk is treated as a "reaction layer" with the same spatial dimension as the membrane.

REVIEWERS' COMMENTS:

Reviewer #3 (Remarks to the Author):

I think the authors have properly replied to my comments and the comments from other reviewers. The revised manuscript contains reasonable data to justify the authors' claim. The authors have also clarified and summarised the current position of understanding the system. This work does not complete our understanding: quantitative comparison between the model and experiments, and the lack of intuition for downstream propagation. Still, I believe this study initiates future works for the complete understanding of the Min system under flow and, more importantly, the application to other systems. Therefore, I would recommend this manuscript for publication in Nature Communications.